

# Technical Note: A Technique to Convert $NO_2$ to $NO_2^-$ with S(IV) and its Application to Measuring Nitrate Photolysis

Aaron Lieberman, Julietta Picco, Murat Onder, and Cort Anastasio

Department of Land, Air, and Water Resources, University of California - Davis, Davis, California 95616, United States

Corresponding author: Cort Anastasio, e-mail: canastasio@ucdavis.edu

**Abstract**

Nitrate photolysis is a potentially significant mechanism for "renoxifying" the atmosphere, i.e., converting nitrate into nitrogen oxides (nitrogen dioxide ($NO_2$) and nitric oxide (NO)) and nitrous acid (HONO). Nitrate photolysis in the environment occurs through two channels, which produce: (1) $NO_2$ and hydroxyl radical ($^\bullet OH$) and (2) nitrite ($NO_2^-$) and an oxygen atom ($O(^3P)$). Although the aqueous quantum yields and photolysis rate constants of both

channels have been established, field observations suggest that nitrate photolysis is enhanced in the environment. Laboratory studies investigating these enhancements typically only measure one of the two photo-channels, since measuring both channels generally requires separate analytical methods and instrumentation. However, measuring only one channel makes it difficult to assess whether secondary chemistry is enhancing one channel at the expense of the other, or if there is an overall enhancement of nitrate photochemistry. Here, we show that the addition of

S(IV), i.e., bisulfite and sulfite, can convert $NO_2$ to $NO_2^-$, allowing measurement of both nitrate photolysis channels with the same equipment. By varying the concentration of S(IV) and exploring method parameters, we determine the experimental conditions that quantitatively convert $NO_2$ and accurately quantify the resulting $NO_2^-$. We then apply the method to a test case, showing how an $^\bullet OH$ scavenger in solution prevents the oxidation of $NO_2^-$ to $NO_2$ but does not enhance the overall photolysis efficiency of nitrate.


## 1.0 Introduction

Nitrogen oxides (i.e., nitrogen dioxide ($NO_2$) and nitric oxide (NO)) and nitrous acid (HONO) are reactive species that play key roles in the formation of tropospheric ozone and hydroxyl radicals ($^\bullet OH$) (Acker et al., 2006;

Seinfeld and Pandis, 2006). The fast oxidation of $NO_2$ to $HNO_3$ is an important sink of gas-phase $NO_x$, while the resulting gas-phase nitric acid and aqueous nitrate are traditionally considered relatively stable reservoir species (Stavrakou et al., 2013; Ye et al., 2017). Although nitrate can photolyze to reform $NO_x$, the lifetime of nitrate is long enough that the small production rates of $NO_x$ and HONO from nitrate photolysis have been considered important only in remote areas (Romer et al., 2018). However, field studies over the past several decades have often

shown that atmospheric measurements of HONO and $NO_x$ are higher than modeled values (Li et al., 2014; Zhou et



al., 2002; Romer et al., 2018). This suggests that nitrate photolysis in the atmosphere is faster than originally considered and, therefore, might be a more significant source of HONO and $NO_x$ (Kasibhatla et al., 2018; Andersen et al., 2023; Zhou et al., 2003).

In sunlight (i.e., for wavelengths above 280 nm) aqueous $NO_3^-$ photolysis proceeds through two channels (Figure 1): the first channel produces $NO_2$ and $^\bullet O^-$ (which is rapidly protonated to $^\bullet OH$) and the second produces nitrite ($NO_2^-$) and an oxygen atom $O(^3P)$. Channel 1 has an average quantum yield of $(1.19 \pm 0.29)\%$ at 293 K for illumination wavelengths above 300 nm (Chu and Anastasio, 2003; Zellner et al., 1990; Warneck and Wurzinger, 1988; Zepp et al., 1987), as shown in Table S1. The quantum yield for channel 2 is sometimes erroneously reported to be an order of magnitude smaller than that of channel 1, but in fact the values are comparable: channel 2 has an

average quantum yield of $(0.98 \pm 0.11)\%$ at 293 K for wavelengths above 300 nm (Benedict et al., 2017; McFall et al., 2018; Warneck and Wurzinger, 1988; Goldstein and Rabani, 2007) (Table S1).

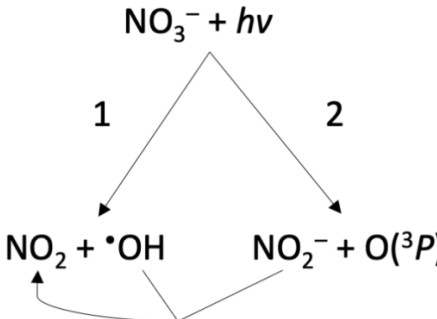

**Figure 1.** The two primary reactions (1 and 2) occuring during nitrate photolysis under tropospheric wavelengths.

$^\bullet OH$ in channel 1 is produced from the protonation of $^\bullet O^-$, which is the primary photoproduct. The unlabeled arrows shows the $^\bullet OH$-mediated oxidation of $NO_2^-$ to $NO_2$, which is a secondary reaction that alters the production rates and quantum yields of $NO_2$ and $NO_2^-$.

These two quantum yields have been determined using different analytical methods. Generally, researchers either monitor the production of hydroxyl radical ($^\bullet OH$) from channel 1 or the production of nitrite ($NO_2^-$) from channel 2. $^\bullet OH$ is typically quantified using a chemical probe (e.g., benzoic acid) that reacts to form a stable product (e.g., *p*-hydroxybenzoic acid) that is monitored by HPLC (Chu and Anastasio, 2003). In contrast, $NO_2^-$ is typically measured via ion chromatography or the more sensitive longpath-Griess method that derivatizes nitrite and measures the highly colored azo-product (Benedict et al., 2017; Ridnour et al., 2000).

Other studies have measured the gas-phase production of $NO_2$ and/or HONO, which is formed from the protonation of $NO_2^-$. However, these gas-phase studies are limited to a specific pH range in order to measure HONO production (Scharko et al., 2014), employ separate instruments to measure HONO and $NO_2$, and focus on how the production rates of $NO_2$ and HONO depend on experimental conditions (Frey et al., 2013; Ma et al., 2021; Liang et





al., 2021).  Although it is possible to measure both $NO_x$ and HONO with commercially available instruments,

researchers often engineer their own instrument to measure HONO and operate a second analyzer for the $NO_2$

channel (Shi et al., 2021; Wang et al., 2021a; Ma et al., 2021).  Furthermore, gas-phase studies do not measure

quantum yields, but instead examine how the production rates of $NO_2$ and/or HONO are altered by factors such as

the presence of other chemical species.

Typically researchers define an enhancement in nitrate photolysis as an experimentally measured

production rate or quantum yield divided by the value under a standard condition (Liang et al., 2021; Wang et al.,

2021b; Shi et al., 2021; Zhou et al., 2003).  For example, a measured apparent nitrite quantum yield of 8% in the

presence of light-absorbing vanillic acid (Wang et al., 2021b) represents an 8-fold enhancement.  If we want to fully

understand the impact of an enhancement, the quantum yields for both channels must be measured.  For instance, if

one measures only the $NO_2$ channel and discovers an enhanced formation rate, it would be unclear whether $NO_2^-$

production also increased or if $NO_2^-$ is being converted to $NO_2$.  Therefore, it would be useful to be able to measure

both channels of nitrate photolysis using a single analytical method.

One possible method to measure both channels is by reducing $NO_2$ to $NO_2^-$ after photolysis, such that the

total measured $NO_2^-$ is the combination of $NO_2$ from channel 1 and $NO_2^-$ from channel 2.  S(IV) (i.e., sulfite ($SO_3^{2-}$)

and bisulfite ($HSO_3^-$)) can reduce $NO_2$ to $NO_2^-$ through the following overall reaction (Lee and Schwartz, 1982;

Clifton et al., 1988; Wang et al., 2020; Song et al., 2021):

$$2\ NO_2 + HSO_3^- + H_2O \rightarrow 2\ NO_2^- + 3\ H^+ + SO_4^{2-}. \tag{R1}$$

Although industry has used this reaction to convert $NO_2$ to $NO_2^-$, they often operate at very high temperatures, or

include additives to enhance the diffusion of $NO_2$ into the aqueous phase (Shen and Rochelle, 1998; Lian et al.,

2022).

Our goal is to use S(IV) chemistry to determine both channels of nitrate photolysis by performing two

experiments using the same analytical method.  In the first run, we measure $NO_2^-$ production directly to quantify

channel 2.  In the second experiment, we use S(IV) to convert photoproduced $NO_2$ to $NO_2^-$ so that the measured

nitrite represents the sum of both $NO_2$ and $NO_2^-$.  Then we quantify channel 1 by subtracting the $NO_2^-$ experiment

result from the combined ($NO_2 + NO_2^-$) experiment result.  If this approach is successful, it would simplify and

expand our ability to analyze $NO_2$ and $NO_2^-$.

**2.0 Methods**

**2.1 Materials**

Information about materials and chemicals is in Section S1 of the supplement.

**2.2 Sample Illumination**

Illumination solutions were prepared daily, were air saturated, and contained 50 µM $NaNO_3$, either 0 or 50

µM 2-propanol, and varying concentrations of S(IV).  The pH of the solution was either controlled by a 0.010 M

phosphate buffer or the added S(IV).  Samples were illuminated with 313 nm light from a 1000 W Hg/Xe arc lamp





with a downstream monochromator (Spectral Energy) and a 310 nm long-pass filter upstream of the sample. 800 μL of aqueous sample in an upright 2 mL HPLC vial (low impurity Type I Class A borosilicate glass, 12 mm O.D. × 32 mm H, Shimadzu) sealed with a septum cap was illuminated from its side. Samples were illuminated with constant stirring in a custom-built, Peltier-cooled aluminum housing (Paige Instruments) that was held at 20 °C by a

recirculating water bath. Samples were kept sealed throughout the illumination. Dark controls containing the same solution as the illuminated sample but not exposed to light were analyzed periodically throughout each experiment. Nitrite production was never detected in the dark controls. Under our conditions, experiments without S(IV) produced no more than 180 nM $NO_2^-$, and experiments with S(IV) produced no more than 180 nM $NO_2 + NO_2^-$.

**2.3 Measurement of Nitrite**

After illuminating all the samples for a given experiment, we determined nitrite concentrations using the Griess method, a spectrophotometric technique that forms an azo-dye complex (Doane and Horwath, 2003; Benedict et al., 2017; McFall et al., 2018). Our experiments had three different sample treatments: (1) no S(IV) in solution, (2) S(IV) in solution during illumination, and (3) S(IV) added to the solution after illumination. Each treatment

required a slightly different method to efficiently form the azo-dye.

For samples without S(IV), the Griess method (Pratt et al., 1995; Moorcroft et al., 2001; Ridnour et al., 2000; Benedict et al., 2017) could be used without adaptation. Within 10 minutes of stopping illumination, we added 25 μL of a 1% sulfanilamide in 10% HCl (v/v) solution, and let it react for 10 minutes in the dark. We then added 25 μL of 0.1% N-(1-Naphthyl)ethylenediamine dihydrochloride (NED) solution and allowed it to react for 10

minutes to form the azo-dye.

Treatment 2, where S(IV) was present in solution during illumination, required an additional step because S(IV) interferes with the Griess reagents (SI Section S2). After illumination, we first added hydrogen peroxide ($H_2O_2$) to the 800 μL sample to obtain a 2:1 molar ratio of $H_2O_2$:S(IV). This was done to oxidize S(IV) to sulfate, which does not interfere with nitrite determination. Within one minute of adding $H_2O_2$, we added 50 μL of 1%

sulfanilamide in 30% (v/v) HCl solution and allowed the solution to react for 10 minutes in the dark. Then we added 50 μL of 0.1% NED solution to the sample and allowed it to react for another 10 minutes in the dark.

Treatment 3 is similar to treatment 2 with one key difference: S(IV) was added to the solution after illumination. Because $NO_2$ is volatile and would escape the illumination container if opened, we developed a method to add the S(IV) without opening the vial. This was done by using a syringe with a hypodermic needle to

directly inject 37.5 μL of a 33.3 mM sulfite solution at pH 9 through the septum into the HPLC vial immediately after the illumination was stopped. The vial was then left to react while stirring for 30 minutes in the dark at room temperature to completely convert $NO_2$ to $NO_2^-$. The samples were then treated exactly as in treatment 2, adding $H_2O_2$, then 50 μL of sulfanilamide in 30% HCl and then 50 μL of NED.

Once the azo-dye was formed, we measured light absorption at ~540 nm in the developed solutions using a

TIDAS II spectrophotometer (World Precision Instruments) with a liquid waveguide capillary cell (LWCC; nominal length of 100 cm, effective path length of 94 cm, 250 μL volume), and a tungsten lamp. The TIDAS contains two lamps, but the deuterium lamp (200−350 nm) caused an artifact in previous experiments (Benedict et al., 2017), so it



was kept off during our measurements. The absorption spectrum was measured from 350 to 700 nm so we could correct for any baseline shifts. The peak height between 530 and 550 nm was determined as the difference between

the maximum absorbance in this range relative to a baseline drawn from the local absorption minima between 400 and 500 nm and between 550 and 700 nm. The limit of detection for nitrite was 7 nM. Fresh standards of sodium nitrite (0 to 200 nM) were prepared daily and used to calibrate the spectrophotometer. As S(IV) and $H_2O_2$ decrease the absorbance response from the spectrophotometer (Figure S2), the same concentrations of S(IV) and $H_2O_2$ used in the samples were also added to the standards to correct for this matrix effect. Samples and other solutions were

manually injected into the LWCC with a syringe, and 4 mL of Milli-Q water was injected between samples to eliminate carry over. We cleaned the LWCC both before and after each experiment with 1 mL injections of three separate cleaning solutions: 1 M NaOH, 1 M HCl, and 50% methanol/50% Milli-Q (MQ) water, with pure MQ injected between each cleaning solution.

Daily controls included a replicate standard, MQ injection as a check for carry over, and a secondary check

standard (Dionex). Analyses were deemed acceptable if the MQ check was below the lowest non-zero standard (10 nM $NO_2^-$) and if both the replicate standard and secondary check standard concentrations were within 15% of known values.

### 2.3 Chemical Actinometry and Calculation of Quantum Yield

The photon flux was measured daily using 2-nitrobenzaldehyde (2NB) as a chemical actinometer (Galbavy et al., 2010). Actinometry was performed under the same conditions (container, volume of sample, temperature) as nitrate photolysis. Under low-light-absorbing conditions, the measured rate constant for 2NB loss during 313 nm irradiation ($j_{2NB,313}$) is calculated using


$$j_{2NB,313} = 2.303 \; x \; 10^3 (I_{313}l)(\varepsilon_{2NB,313}\phi_{2NB,313}), \qquad (1)$$

where $I_{313}l$ is the surface-area-normalized photon flux (mol-photon cm$^{-2}$ s$^{-1}$), $\varepsilon_{2NB,313}\phi_{2NB,313} = 640$ M$^{-1}$ cm$^{-1}$ is the product of the base-10 molar absorption coefficient and quantum yield for 2NB at 313 nm (Anastasio et al., 1994),

2.303 converts $\varepsilon$ to base-e, and $10^3$ cm$^3$ L$^{-1}$ is for units conversion. Similarly, the formation rate constant for nitrite from nitrate photolysis is:

$$j(NO_3^- \rightarrow NO_2^-)_{313} = \; 2.303 \; x \; 10^3 (I_{313}l)\left(\varepsilon_{NO_3^-,313}\right)\left(\phi(NO_2^-)_{313}\right), \qquad (2)$$

where $\phi(NO_2^-)_{313}$ is the quantum yield of nitrite formation from nitrate photolysis at 313 nm, and $\varepsilon_{NO_3^-,313}$ is the base-10 molar absorption coefficient of nitrate at 313 nm, 5.29 M$^{-1}$ cm$^{-1}$ (Chu and Anastasio, 2003).

The rate of nitrite formation from nitrate photolysis, d[$NO_2^-$]/dt, is a first-order process:

$$\frac{d[NO_2^-]}{dt} = j(NO_3^- \rightarrow NO_2^-)_{313}[NO_3^-]. \qquad (3)$$



Since the experiments were at short time scales where nitrate loss was negligible, the increase of nitrite was linear, and the nitrite formation rate could be determined with simple linear regression. Combining equations 1-3 allows us to solve for the quantum yield of nitrite:

$$\phi(\mathrm{NO_2^-})_{313} = \frac{\mathrm{d[NO_2^-]}}{\mathrm{dt}} \: x \: \frac{\varepsilon_{2NB,313}\phi_{2NB,313}}{j_{2NB,313}\varepsilon_{NO_3^-,313}[NO_3^-]}.$$

(4)

For simplicity, and because all our experiments were performed with 313 nm illumination, we omit the "313" subscript throughout the rest of this manuscript.

**2.4 Combined Quantum Yield and $\phi(\mathrm{NO_2})$ Calculations**

For experiments with added S(IV), the measured concentration of nitrite represents both the primary nitrite from nitrate photolysis as well as secondary nitrite formed by conversion of $NO_2$. Thus, the calculated quantum yield in experiments with S(IV), i.e., $\phi(\mathrm{NO_2^-})_{S(IV)}$, is a combination of the quantum yields for both channels 1 and 2:

$$\phi(\mathrm{NO_2^-})_{S(IV)} = \phi(\mathrm{NO_2^-}) + f \times \phi(\mathrm{NO_2}).$$

(5)

Here $f$ is the fraction of $NO_2$ that reacts with S(IV) to make $NO_2^-$, as opposed to going down other pathways:

$$f = \frac{k_{\mathrm{HSO_3^-+NO2}} \times [HSO_3^-] + k_{\mathrm{SO_3^{2-}+NO2}} \times [SO_3^{2-}]}{k_{\mathrm{HSO_3^-+NO2}} \times [HSO_3^-] + k_{\mathrm{SO_3^{2-}+NO2}} \times [SO_3^{2-}] + k_{\mathrm{other}}},$$

(6)

where $k_{\mathrm{S(IV)+NO2}}$ is the reaction rate constant of S(IV) and $NO_2$, $1.2 \times 10^7$ M$^{-1}$ s$^{-1}$ and $1.7 \times 10^7$ M$^{-1}$ s$^{-1}$ for bisulfite and sulfite, respectively (Clifton et al., 1988) and $k_{\mathrm{other}}$ is the pseudo-first-order rate constant for all other pathways that consume $NO_2$. The concentrations of bisulfite and sulfite are determined based on the total S(IV) in solution, [S(IV)], and their mole fractions, which depend on the two p$K_a$ values for S(IV) (p$K_{a1}$ = 1.9, p$K_{a2}$ = 7.2; Seinfeld and Pandis (2006)). As described below, at pH ≈ 8 a S(IV) concentration of 1.5 mM and higher is sufficient to make $f$ equal 1, i.e., S(IV) is essentially the only fate for $NO_2$, so it is quantitatively converted to $NO_2^-$. Under this condition, we calculate the quantum yield for $NO_2$ formation, $\phi(\mathrm{NO_2})$, as the difference between the measured nitrite quantum yields in the presence and absence of S(IV):

$$\phi(\mathrm{NO_2}) = \phi(\mathrm{NO_2^-})_{S(IV)} \text{ - } \phi(\mathrm{NO_2^-}).$$

(7)

**3.0 Results**

**3.1 Modification of the Griess Method for Solutions Containing S(IV)**





As described in Section S2, we found that the addition of micromolar levels of S(IV) interferes with the determination of nitrite because of two issues: (1) it prevents the formation of the azo-dye derivative and (2) it moves the solution acidity out of the required range. The first issue was solved by oxidizing the S(IV) to S(VI) with $H_2O_2$ prior to the addition of the Griess reagents (Figure S1). We added $H_2O_2$ to the samples such that there was a 2:1 molar ratio of $H_2O_2$:S(IV), then within 1 minute of the addition of $H_2O_2$, we added the sulfanilamide solution

and, 10 min later, the NED reagent. After waiting another 10 minutes, we measured the UV-VIS spectra for the entire batch of samples within 20 minutes of capturing the spectrum of the first sample. We also doubled the standard volumes of both Griess reagents added to the sample solutions to ensure that there were enough reactants to form the azo-dye.

       The second issue caused by S(IV) was that it pushed the solutions to pH 7. This basicity prevented the

conversion of nitrite to the azo-dye because this reaction requires a pH below 2. The standard 10% HCl (v/v) in the sulfanilamide solution only lowered the sample pH to ~ 4 for solutions containing 1.5 mM of sulfite. Per the recommendation by Doane and Horwath (2003), we increased the HCl concentration in the sulfanilamide solution to 30% (v/v), which lowered the pH of the sample-sulfanilamide mixture to less than 2, overcoming the pH issue caused by S(IV).


**3.2 Addition of S(IV) to Solution Prior to Illumination**

       Our goals in this initial set of experiments were to examine whether S(IV) in solution can convert photoproduced $NO_2$ to $NO_2^-$ and, if so, to determine the concentration of aqueous S(IV) required to make this conversion quantitative, i.e., close to 100%. If S(IV) can quantitatively convert $NO_2$ to $NO_2^-$, then the measured

nitrite quantum yield at this S(IV) concentration should equal the sum of the quantum yields from both channels of nitrate photolysis.

       We started experiments by running a test without S(IV) (50 µM NaNO₃, 50 µM 2-propanol, and 293 K) to confirm that our result matches the literature. The average $\phi(NO_2^-)$ from our four replicate experiments without S(IV) is (1.05 ± 0.06)%, which is statistically no different ($p = 0.36$) from the average of the literature values shown

in Table S1, (0.98 ± 0.11)%. Then we began performing experiments with increasing concentrations of S(IV). As [S(IV)] increases, the apparent nitrite quantum yield increases until it reaches a plateau for S(IV) concentrations at roughly 500 µM and above (Figure 2). The measured quantum yield at the plateau, determined as the average (± 1 σ) of the individual experiments from 500 to 2000 µM S(IV), is (2.01 ± 0.05)%. This is slightly lower than, but statistically indistinguishable ($p = 0.14$) from the sum of the average literature quantum yields for both channels,

(2.17 ± 0.52)%, which is shown as the upper horizontal line in Figure 2. We then calculate $\phi(NO_2)$ by taking the difference between the quantum yield determined with S(IV), which measures the sum of the two channels, and the quantum yield for the nitrite channel (Eq. 7). This results in a value of $\phi(NO_2)$ of (0.96 ± 0.12)%, which is slightly lower than the average of previous experiments (1.19 ± 0.29)%, but statistically no different ($p = 0.10$). These results confirm that S(IV) in the reaction solution during illumination can quantitatively convert photochemically

produced $NO_2$ to $NO_2^-$, allowing the Griess spectrophotometric technique to quantify both channels of nitrate photolysis.





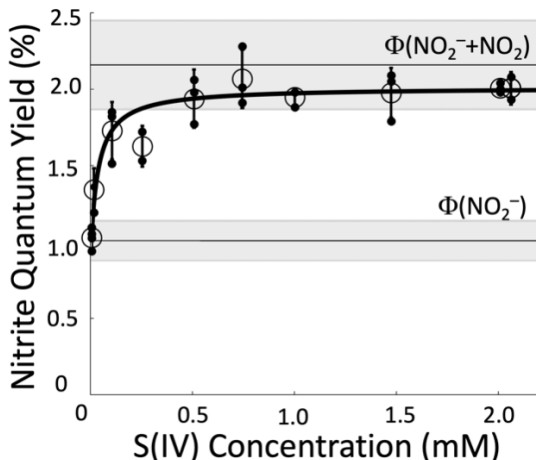

**Figure 2.** Measured apparent nitrite quantum yields for the photolysis of 50 µM nitrate solutions (293 K, pH 8) in
the presence of different concentrations of S(IV). Hollow circles represent the average (± 1 σ) of individual
experiments, which are shown as solid black points. The line through the data is a fit to equation 8. The lower grey
area centered at 1.1% is the average of previously determined values of $\phi(NO_2^-) \pm 1\ \sigma$, and the upper grey area
centered at 2.2% is the sum of the quantum yields from both channels, $\phi(NO_2^- + NO_2) \pm 1\sigma$, from the literature.
Literature values used to calculate these averages are in Table S1.


We can also use our Figure 2 data to estimate the value for $k_{other}$, the pseudo-first-order rate constant for
$NO_2$ loss due to other pathways, i.e., not reacting with S(IV). Combining equations 5 and 6 yields:

$$\phi(NO_2^-)_{(SIV)} = \phi(NO_2^-) + \phi(NO_2) \times \frac{k_{HSO_3^- + NO_2} \times [HSO_3^-] + k_{SO_3^{2-} + NO_2} \times [SO_3^{2-}]}{k_{HSO_3^- + NO_2} \times [HSO_3^-] + k_{SO_3^{2-} + NO_2} \times [SO_3^{2-}] + k_{other}}.$$

(8)

Fitting this equation to our data using Python (Van Rossum and Drake Jr, 1995) yields the solid line in Figure 2 and
parameter values of $k_{other} = 700 \pm 300$ s$^{-1}$, $\phi(NO_2) = (0.94 \pm 0.07)\%$, and $\phi(NO_2^-) = (1.10 \pm 0.06)\%$. We can use the
value of $k_{other}$ in equation 6 to calculate the percent of $NO_2$ that is converted to $NO_2^-$ in solutions at a given S(IV)
concentration and pH value: values are 96%, 98%, and 99% at 500, 1000, and 1500 µM S(IV), respectively, at pH 8.

### 3.3 Addition of S(IV) After Illumination

Our experiments above used S(IV) in the illumination solution to convert $NO_2$ to $NO_2^-$. While this method
works, it has the disadvantage that S(IV) might interfere with other reactive species or reaction pathways during
illumination. To avoid this problem, in this section we examine whether we can prevent $NO_2$ from escaping the
sample container and convert it to nitrite by adding S(IV) to the solution after illumination.



We made several changes to the procedure in Section 3.2 to ensure full conversion of $NO_2$ to $NO_2^-$ when adding S(IV) after illumination. We examined the effectiveness of the potential changes based on a single trial where we tested three different treatments of the samples post-illumination: (1) adding 1.5 mM S(IV) to the samples

and allowing them to stir for 30 minutes in the dark, (2) adding 1.0 mM S(IV) and stirring for 30 minutes, and (3) adding 1.5 mM S(IV) and stirring for 10 minutes. In each case, we added the S(IV) immediately after the end of sample illumination by injecting a small volume, 25 or 38 µL, of a 33 mM sodium sulfite stock solution through the septum of the HPLC cap with a syringe and hypodermic needle. The goal with this technique as to keep the illumination container sealed so that $NO_2$ could not escape. Measured values of $\phi(NO_2^-)_{S(IV)}$ were $(1.97 \pm 0.24)\%$,

$(1.53 \pm 0.19)\%$, and $(1.60 \pm 0.45)\%$ for treatments 1, 2, and 3, respectively. The only trial that seemed to completely convert all the $NO_2$ to $NO_2^-$ was the first treatment, i.e., 1.5 mM S(IV) with 30 min of stirring. As such, we used this treatment method going forward.

We also estimated the timescale of $NO_2$ conversion to nitrite to compare with our experimental results. Based on the volumes in the reaction vial (800 µL of solution and ~1.3 mL of headspace), Henry's Law predicts (at

293 K) that 10% of $NO_2$ should be in the aqueous phase and 90% in the head space. Based on the kinetic data from Clifton et al. (1988), the lifetime of total $NO_2$ in the vial is approximately 1 ms. This means that there should have been no difference between the results of trial 1 and 3, which is not the case. It is unclear why there is a discrepancy between the theoretical and experimental timescales for the conversion of $NO_2$ to $NO_2^-$.

Next, we examined whether the addition of S(IV) after illumination produced results that were the same as

those for experiments where S(IV) was in the solution during illumination. We performed triplicate experiments measuring the combined $NO_2 + NO_2^-$ quantum yield in pH 5 solution containing 50 µM $NaNO_3$ and 50 µM 2-propanol, and 1.5 mM S(IV) added to the solution after illumination. As shown in Figure S4, the average $\pm 1\sigma$ combined quantum yield from this set of experiments is $(2.10 \pm 0.08)\%$. This is statistically no different from the result we obtained above when S(IV) was present in the solution during illumination, $(2.00 \pm 0.14)\%$ ($p = 0.32$), and

no different from the literature value, $(2.17 \pm 0.52)\%$ ($p = 0.74$; Table S1). This indicates that we can add S(IV) after the photoproduction of nitrogen dioxide has stopped and still convert all the $NO_2$ to $NO_2^-$.

### 3.4 Applying the S(IV) Method: Impact of an $^\bullet OH$ Scavenger

Our final step is to show the utility of determining both $NO_2$ and $NO_2^-$ in a chemical system, by using the

example of quantifying the impact of an $^\bullet OH$ scavenger on the two channels from nitrate photolysis. Based on past work (Benedict et al., 2017; Roca et al., 2008; McFall et al., 2018), in the absence of a hydroxyl radical scavenger, we expect that $^\bullet OH$ will react with $NO_2^-$ to form $NO_2$ (Figure 1). In this no-scavenger case, the $NO_2^-$ quantum yield should be underestimated and the $NO_2$ quantum yield should be overestimated by an equal amount. In contrast, adding a scavenger suppresses the hydroxyl radical concentration and its impact on both photoproducts. However,

we expect that the combined quantum yield, i.e., the sum of values for both channels, will be the same regardless of the presence of $^\bullet OH$ scavengers. That is, we expect that an $^\bullet OH$ scavenger will prevent the conversion of $NO_2^-$ to $NO_2$ but will not alter the overall photochemical efficiency of nitrate photolysis. While the impact of $^\bullet OH$





scavengers on the nitrite channel has been examined previously, we are unaware of any past attempts to measure both channels in the presence and absence of scavengers.


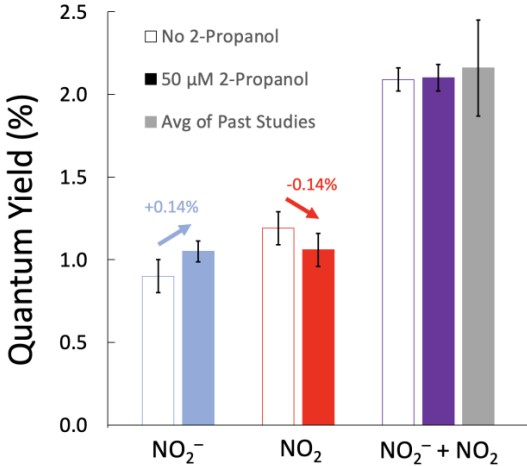

**Figure 3.** Measured quantum yields of nitrite (yellow bars), nitrogen dioxide (red bars), and both products (purple bars) from the photolysis of 50 µM $NO_3^-$ at 293 K and pH 5. The hollow bars are experiments without 2-propanol (an $^\bullet OH$ scavenger), while solid bars represent experiments with 50 µM 2-propanol. The grey bar is the sum of the

average quantum yields for the two channels from past studies (Table S1). Arrows above the $NO_2^-$ and $NO_2$ channels indicate the impact of the $^\bullet OH$ scavenger. Error bars are ± 1σ. Errors for the $NO_2^-$ and ($NO_2^-$ + $NO_2$) quantum yields were calculated from replicate experiments; these errors were propagated to determine the error for the $NO_2$ channel result.

320         As shown by the blue arrow in Figure 3, the addition of an $^\bullet OH$ scavenger increases the $NO_2^-$ quantum yield (by 0.14%), as expected since it impedes the oxidation of nitrite by hydroxyl radical (Figure 1). Also consistent with our model above, the red arrow shows that the $^\bullet OH$ scavenger decreases the quantum yield of the $NO_2$ channel (by 0.14%), a result of the suppression of nitrite oxidation by $^\bullet OH$ to make $NO_2$. The $NO_2^-$ quantum yields without S(IV), with and without 2-propanol, are statistically different ($p = 0.04$). However, when S(IV) is

added to solution, the presence or absence of an $^\bullet OH$ scavenger has no impact ($p = 0.95$) on the sum of the quantum yields for the two channels. This is what we expect because the $NO_2$ that was formed from the reaction of $^\bullet OH$ and $NO_2^-$ is converted back to $NO_2^-$ by S(IV), resulting in the same total amount of $NO_2^-$ + $NO_2$ in the two sets of experiments. This shows that the addition of a $^\bullet OH$ scavenger does not impact the overall efficiency of nitrate photolysis (i.e., the sum of the quantum yields of the two channels) but prevents the oxidation of $NO_2^-$ to $NO_2$.

330         Our quantum yields in this set of experiments are in good agreement with previously determined values. As mentioned in section 3.2, our nitrite quantum yield without S(IV) agrees with previously reported $\phi(NO_2^-)$ values. Our combined quantum yield values, $\phi(NO_2^-)_{S(IV)}$, are (2.10 ± 0.08)% and (2.09 ± 0.16)% with and without



an ˙OH scavenger, respectively (Figure 3). Our values here agree with the Table S1 average of previously determined combination of both channels, $(2.17 \pm 0.52)\%$ ($p > 0.70$). The $NO_2$ channel was calculated, using Eq. (7), as the difference in the quantum yield between experiments with S(IV) added after illumination and experiments without S(IV). In this set of experiments, our measured $\phi(NO_2)$, $(1.05 \pm 0.10)\%$, is similar to the average of the literature values, $(1.19 \pm 0.29)\%$ ($p = 0.47$), as shown in Table S1.

**4.0 Impacts/Implications**

We have demonstrated that S(IV) can convert aqueous $NO_2$ to $NO_2^-$, which allows the production of both the gas- and aqueous-phase products of nitrate photolysis to be quantified in the aqueous phase in a sealed container using the same analytical method. Although nitrate photolysis is traditionally considered a minor source of $NO_x$, recent research has shown that the efficiency of nitrate photolysis can be enhanced by other light-absorbing compounds or its physical environment (Wang et al., 2021b; Mora Garcia et al., 2021; McFall et al., 2018). It is important to understand whether an apparent enhancement impacts only one channel, both channels, or is due simply to a conversion of one product to another. As many field studies have noted that the measured enhancement impacts the $NO_2^-$ channel more than the $NO_2$ channel, it is likely that different chemicals impact nitrate photolysis in a variety of ways (Andersen et al., 2023; Kasibhatla et al., 2018; Ye et al., 2016). Understanding how different variables impact nitrate photolysis will allow a more comprehensive understanding of nitrogen cycling and should improve model predictions of ambient $NO_x$ and HONO concentrations. Performing experiments with and without S(IV) for a given experimental condition will allow laboratory experiments to determine both channels of nitrate photolysis, which will reveal whether one or both channels are enhanced or if secondary chemistry is converting one product to the other.

This new S(IV) method also has applications beyond nitrate photolysis, as it can be used for any system where $NO_2$ needs to be quantified. This could include studies where $NO_2$ production occurs in the aqueous phase, such as the decomposition of metallic nitrate compounds (Gallagher et al., 1971; Yuvaraj et al., 2003), or in studies where the production of $NO_2$ is small enough that it cannot be quantified by commercially available analyzers.

**Code and data availability.**

All data and code can be obtained by emailing the corresponding author at canastasio@ucdavis.edu

**Author Contributions**

AL and CA conceptualized the study. AL, JP, and MO performed the experiments in the study. AL performed the coding and data analysis. AL wrote the paper with input from CA.

**Competing interests**

The authors declare that they have no conflict of interest.



## 370     Financial support

This research was supported by the National Science Foundation, grant CHE-2305164.



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
