# Peer review of "Technical Note: A Technique to Convert $\text{NO}_2$ to $\text{NO}_2^-$ with S(IV) and its Application to Measuring Nitrate Photolysis"

_EGUsphere, 2023_

## Author Response (AR2)

Response to Reviewers for "Technical Note: A Technique to Convert $NO_2$ to $NO_2^-$ with S(IV) and its Application to Measuring Nitrate Photolysis" by Lieberman et al.

Each reviewer comment is listed in italics and our response, in plain text, is directly below it. Line numbers in the revised version are different from the original (e.g., in a reviewer's comment) due to changes in the manuscript.

*Anonymous Referee #1*

*The manuscript by Lieberman et al. details a method to measure dissolved nitrogen dioxide ($NO_2$) concentrations in aqueous solutions. The method relies on the conversion of aqueous $NO_2$ into nitrite using S(IV) (= $HSO_3^-$ + $SO_3^{2-}$), followed by quantitation of nitrite by the Griess method. This technical note is well-written and provides enough detail to easily reproduce the experiments. The method is useful and represents a novel approach to (near)simultaneously measuring nitrite and $NO_2$ concentrations arising during nitrate photolysis. The application of this method to studying nitrate photoproduct yields is notable given the importance of elucidating renoxification pathways derived from nitrate photolysis. As pointed out by the authors, previous measurements have been limited to measuring the quantum yields of the nitrite and $NO_2$ photolysis channels with different analytical methods. The current method will enable the use of a single method to assess whether secondary chemistry is enhancing one channel at the expense of the other or if there is an overall enhancement of the primary nitrate quantum yield. Below are some points that came up during my reading of the manuscript.*

*Figure 1: The content of this figure / scheme really doesn't lend itself well to its own figure. I feel it would be better to simply replace this with three separate equations (R1-R3) in the main text.*

Thank you for the suggestion. We replaced the figure with separate equations, R1 and R2 in section 1, and R4 in section 3.4.

*Line 99: Please specifically indicate the pH (or pH range) of the solution here.*

We have edited the section to describe that the phosphate buffer was either pH 8 or 5.

*Section 2.3: Please address the possibility that $HSO_3^-$ can act as a scavenger of OH radical produced from nitrate photolysis. Also, this reaction would form sulfite radical anion and I would like to know your thoughts on whether this would interfere with the chemistry. Does one have to correct the quantum yields to account for this? In addition to this, please address the possibility that $H_2O_2$ can oxidized the Griess reaction reagents or nitrite, converting it to nitrate via peroxynitrite. I would expect for a paper such as this for the researchers to test potential interferences with the method. This is not done beyond the limited intercomparison of nitrite and $NO_2$ quantum yields presented in the figures. If a more complex matrix was used (e.g., in the presence of organic molecules, dissolved organic matter, transition metals, etc.) could we expect the method to accurately quantitate $NO_2$ concentrations?*

Yes, $HSO_3^-$ will scavenge ˙OH and produce sulfite radical anion, potentially causing interferences. Because of concerns over unwanted chemistry such as this, we developed the method discussed in section 3.3, where we add S(IV) post-illumination, when the hydroxyl radical will no longer be present. Since there was no significant difference between the measured quantum yield for illuminations with S(IV) in solution during illumination and S(IV) added after illumination, we do not believe that either $HSO_3^-$ or sulfite radical impacted the nitrite quantum yield measured in the presence of S(IV).

$H_2O_2$ might interfere at higher concentrations than we used, but we do not see evidence of this with a 2:1 $H_2O_2$:S(IV) ratio. When we tested the ratio of $H_2O_2$:S(IV) we found that the 1:1 and 2:1 ratios produced the steepest calibration curves compared to the 0.6:1 and 3:1 ratios. We speculate that the 0.6:1 ratio has a lower calibration curve slope because of incomplete oxidation of S(IV), which then interfered with the azo-dye formation. The fact that the 3:1 ratio result is less than the 1:1 and 2:1 ratio results might be because $H_2O_2$ at this higher concentration does react with nitrite.

However, even at this highest $H_2O_2$ concentration, oxidation of $NO_2^-$ does not appear fast enough to impact nitrite quantification. Based on the third-order rate constant for the reaction (Lukes et al., 2014), the lifetime of $NO_2^-$ in a pH 5 solution with 100 nM $NO_2^-$ and 3 mM $H_2O_2$ is ~8.5 hours (~6 hour half-life) and the $OONO^-$ formation rate would be 3.3 pM s$^{-1}$. Since the azo-dye formed from the Greiss analysis is generated within 20 minutes of the addition of $H_2O_2$, less than 5% of the nitrite would react with $H_2O_2$ during this time period. This indicates that $H_2O_2$ is not significantly destroying $NO_2^-$ under this condition. We added this information to section S2 of the supplement.

We also investigated the stability of the solution after the addition of the NED reagent. We found that the azo-dye is stable in solution for up to 5 hours after the addition of the NED reagent. After 6 hours, the signal response from the instrument decreased by ~10%. The fact that the signal response only decreased by ~10% compared to the much larger 50% predicted by the $NO_2^-$ half-life, leads us to believe that $H_2O_2$ oxidation of the formed azo-dye is slower than the oxidation of $NO_2^-$ and would have a negligible impact on nitrite determination if the samples are analyzed within 5 hours of adding the $H_2O_2$.

Currently we are working on a research project that uses this technique in solutions with both transition metals and organics. So far, these components have not impacted the method's ability to convert $NO_2$ to $NO_2^-$ or to determine the $NO_2^-$ concentration. However, in all our experiments we match the matrix of our illumination solutions for the nitrite standards. If you do not matrix match the solutions and standards it could lead to improper quantitation. A simple check for matrix effects is an analysis of a spike recovery sample.

*Line 141: How did you determine the limit of detection?*

We realized that the detection limit originally listed in the manuscript was for nitrite without S(IV) in solution. We have determined the detection limit for nitrite with 1.5 mM S(IV) (Armbruster and Pry, 2008) and added this to the manuscript in section 2.3. It is approximately 50% greater than the LOD for solutions without S(IV).

*Lines 210-218:  Practically speaking, can the authors please provide some comments on how variable these results are and how stable the system is.  That is, if I try to use this method, how important is it to follow the indicated timing here?  If one lets the reaction go longer or doesn't "develop" the reaction solutions soon enough, does one get a different answer? Can one store the reaction solutions in the freezer for later analysis and still get comparable answers? Any insights?*

For the purpose of this comment, I will describe the stability of sample treatments 2 and 3 from section 2.3, after the addition of (1) S(IV), (2) $H_2O_2$, and (3) NED to the system.  We added this information to section S3 of the supplement.

After the addition of S(IV), we found that the samples were stable for up to 1 week if kept sealed, covered, and refrigerated.  After the addition of $H_2O_2$, samples were unstable after 5 minutes: after the addition of $H_2O_2$, the reagents for the Greiss method need to be added immediately for results to be precise and accurate.  After the addition of NED, the sample was stable up to 5 hours covered on the benchtop. We did not test the possibility of freezing solutions to extend stability.

*Line 263:  The authors suggest there are other reactions that lead to consumption of and $NO_2$, quantifying it as k$_{other}$. Can the authors please provide some insights into what they think this(ese) reaction(s) is(are)?*

Thank you for the question.  We do not know the specific "other" pathways for $NO_2$ loss in our system.  However, they could include reaction in the solution, reaction on the surface of the container, or potentially evaporation out of the container.

*Anonymous Referee #2*

*This paper presents a method that converts $NO_2$ to $NO_2^-$ using S(IV) so that nitrate photolysis products can be measured by the same instrument. This method can derive total nitrate photolysis quantum yield from the two channels by one analytical method, allowing determination of whether the nitrate photolysis is impacted by one channel or both channels in different environments. This method could potentially be useful to our community in better quantifying nitrate renoxification, but there are issues that need to be addressed.*

*Major comments:*

1. *This method is developed to be used for bulk nitrate solutions. However, many studies (including references cited in this paper) on enhanced nitrate photolysis were about particulate nitrates or nitrates on surfaces. The authors need to address this disconnection. How can this method be used to address particulate nitrate/surface-absorbed nitrate photolysis? With complex composition in atmospheric particles, can this method still be valid*

Thank you for this question. It brings up a valid point about the applicability of this method. We believe the method can be used regardless of whether the photolysis experiment occurs in solution, in particulate matter, or on surfaces. It is possible that chemical components on specific surfaces or in particle samples could interfere with the method. But the amount of S(IV) that we add is large enough that it should be able to overcome most interferences. As an example of the robustness of the method, the experiments performed in section 3.3 show that we can convert not only aqueous $NO_2$ produced during photolysis, but also $NO_2$ that escaped into the gas phase. Therefore, as long as the illumination container is sealed and no $NO_2$ is allowed to escape, the addition of aqueous phase S(IV) to the illumination container should convert $NO_2$ to $NO_2^-$.

1. *Did the authors investigate whether the added $H_2O_2$ would react with $NO_2$- and interfere with the quantum yield quantification?*

As described in our response to question 4 of Reviewer 1, $H_2O_2$ did not interfere with our quantum yield quantification. We now discuss this in Section S2 of the supplement.

*Minor/technical comments*

1. *In equation (1)(2) and (4), the x should be a multiplication sign, not letter x.*

Thank you for pointing out this error. The manuscript has been edited to fix this mistake.

2. *Figure 3 caption: "Measured quantum yields of nitrite (yellow bars)". I think the authors mean blue bars.*

You are correct that the bars were meant to be "nitrite (blue bars)" not yellow. The manuscript has been changed to fix this error.

**References:**

Armbruster, D. A. and Pry, T.: Limit of Blank, Limit of Detection and Limit of Quantitation, Clin Biochem Rev, 29, S49–S52, 2008.

Lukes, P., Dolezalova, E., Sisrova, I., and Clupek, M.: Aqueous-phase chemistry and bactericidal effects from an air discharge plasma in contact with water: evidence for the formation of peroxynitrite through a pseudo-second-order post-discharge reaction of $H_2O_2$ and $HNO_2$, Plasma Sources Sci. Technol., 23, 015019, https://doi.org/10.1088/0963-0252/23/1/015019, 2014.